# A Colorimetric Assay for the Detection of Glucose and H_2_O_2_ Based on Cu-Ag/g-C_3_N_4_/ZIF Hybrids with Superior Peroxidase Mimetic Activity

**DOI:** 10.3390/molecules25194432

**Published:** 2020-09-27

**Authors:** Quan Pan, Yuelin Kong, Kuan Chen, Mi Mao, Xiaohui Wan, Xiaoyan She, Qingsong Gao, Yu He, Gongwu Song

**Affiliations:** 1Hubei Province Fiber Inspection Bureau, Wuhan 430061, China; panquanfq@163.com (Q.P.); yangguangwxh9272@163.com (X.W.); sxy13871171608@163.com (X.S.); gqs0601@163.com (Q.G.); 2Hubei Collaborative Innovation Center for Advanced Organic Chemical Materials, Ministry-of-Education Key Laboratory for the Synthesis and Application of Organic Functional Molecules, College of Chemistry and Chemical Engineering, Hubei University, Wuhan 430062, China; kongyue_lin@163.com (Y.K.); workout_k@163.com (K.C.); xxhh_my@163.com (M.M.); songgw@hubu.edu.cn (G.S.)

**Keywords:** Cu-Ag/g-C_3_N_4_/ZIF hybrid, nanozyme, colorimetric detection, H_2_O_2_, glucose

## Abstract

In this work, we report the synthesis of Cu-Ag bimetallic nanopartiles and g-C_3_N_4_ nanosheets decorated on zeolitic imidazolate framework-8 (ZIF-8) to form a Cu-Ag/g-C_3_N_4_/ZIF hybrid. The hybrid was synthesized and characterized by Transmission electron microscopy (TEM), Fourier transformed infrared (FTIR), the X-ray diffraction (XRD) and X-ray photoelectron spectroscopy (XPS). The Cu-Ag/g-C_3_N_4_/ZIF hybrid has intrinsic peroxidaselike catalytic activity towards the oxidation of TMB in the presence of H_2_O_2_. The situ synthesis of Cu-Ag bimetallic nanopartiles on 2D support such as g-C_3_N_4_ nanosheets would significantly enhance the peroxidaselike catalytic properties of individual Cu-Ag bimetallic nanopartiles and the g-C_3_N_4_ nanosheets. After loading of Cu-Ag bimetallic nanopartiles and g-C_3_N_4_ nanosheets on the ZIF-8, the hybrids exhibited superior peroxidaselike catalytic activity and good recyclability. Then, this method was applied for detecting glucose in human serum, owing the significant potential for detection of metabolites with H_2_O_2_-generation reactions.

## 1. Introduction

Nanozymes have attracted increasing attention in the past decades in the field of catalysis because of their fantastic advantages, including low cost, facile synthesis, excellent stability and are easy to manipulate in optical sensing [1,2]. After the Fe_3_O_4_ nanoparticles was first reported as the peroxidaselike nanoenzyme to catalyze 3,3′,5,5′-tetramethylbenzidine (TMB) in the presence of H_2_O_2_ to produce a blue color compound 3,3′,5,5′-tetramethylbenzidine diimine (TMBDI) [3], different kinds of nanozymes such as carbon based nanomaterials [4], metal nanoparticles [5,6], metal oxides [7,8] and metal oxide nanocomposites [9,10] have been discovered to replace natural enzymes.

Graphitic carbon nitride (g-C_3_N_4_) nanosheets, with similar sheet structure to graphene, have great potential in the catalytic applications of water decomposition, carbon dioxide reduction, organic pollutant degradation and organic synthesis reaction because of their unique band position (2.7 eV) [11,12,13]. However, the catalytic performance of the g-C_3_N_4_ meets some bottlenecks due to its low charge generation efficiency and high photoelectron-hole recombination rate [14]. Therefore, great efforts have been devoted to improve its catalytic activity. Especially, the combination or coupling of the catalyst with oxides [15,16], metals [17], precious metal nanoparticles [18], sulfide [19,20] and carbon materials [21,22] is an effective method to improve the catalytic activity [23].

Herein, we report on the synthesis of Cu-Ag bimetallic nanopartiles and g-C_3_N_4_ nanosheets decorated on zeolitic imidazolate framework-8 (ZIF-8) to form Cu-Ag/g-C_3_N_4_/ZIF nanozyme with intrinsic peroxidaselike catalytic activity towards the oxidation of TMB in the presence of H_2_O_2_ (Scheme 1). Cu-Ag bimetallic nanoparticles have been proved to decorate on reduced graphene oxide nanosheets as peroxidase mimic for glucose and ascorbic acid detection [24]. Thus, situ synthesis of Cu-Ag bimetallic nanopartiles on 2D support such as g-C_3_N_4_ nanosheets would significantly enhance the peroxidaselike catalytic properties of individual Cu-Ag bimetallic nanopartiles and the g-C_3_N_4_ nanosheets. ZIF-8 have been recently investigated as supporters to reduce the consumption rate of metal NPs and restrain their aggregation for its high specific surface area and good thermal and chemical stability [25,26]. After loading of Cu-Ag bimetallic nanopartiles and g-C_3_N_4_ nanosheets on the ZIF-8, the hybrids exhibited superior peroxidaselike catalytic activity and good recyclability. Then, a colorimetric method for H_2_O_2_ and glucose detection assay was reported based on the color change of TMB with the catalytic reaction of glucose with glucose oxidase (GOx).

## 2. Results and Discussion

### 2.1. Characterizations of Cu-Ag/g-C_3_N_4_/ZIF Hybrid

TEM and HRTEM image (Figure 1a,b) showed Cu-Ag bimetallic nanoparticles and was g-C_3_N_4_ formed on the surface of ZIF-8. The average particle size of Cu-Ag bimetallic nanoparticles was about 5 nm. The alloy nature of Cu-Ag/g-C_3_N_4_/ZIF hybrid is confirmed by the appearance of clear lattice fringes in the HRTEM images. The lattice structures of C_3_N_4_ (0.336 nm), Cu NCs (0.207 nm) and Ag NCs (0.23 nm) were observed from Figure 1b, proving that g-C_3_N_4_ nanosheets and Cu-Ag bimetallic nanoparticles were loaded onto ZIF-8. The elemental composition and chemical bonding information on the Cu-Ag/g-C_3_N_4_/ZIF hybrid were studied by XPS analysis (Figure 1c,d). The peaks at 933.1 and 952.6 eV were attributed to Cu 2p, belonging to Cu (0), while the peaks at 368.2 and 374.27 eV were attributed to Ag 3d, belonging to Ag (0), indicating that the Cu (0) and Ag (0) were loaded.

The chemical arrangement of the Cu-Ag/g-C_3_N_4_/ZIF hybrid was analyzed by FTIR (Figure 2a,b). In Figure 2a, the absorption band at 1576 cm^-1^ was associated with the stretching vibration of CN heterocycle. The peak at 891 cm^-1^ could be assigned to the bending vibration of the triazine ring [27]. Compared with bulk-C_3_N_4_, the absorption peak intensity of g-C_3_N_4_ in the infrared spectrum was significantly weakened, indicating that part of the CN heterocyclic ring structure was destroyed when bulk-C_3_N_4_ was stripped into nanosheets. In Figure 2b, the absorption peaks of 3138 and 2933 cm^−1^ were, respectively, the stretching vibration of C-H bond in methyl and imidazole ring, the stretching vibration absorption peaks of C=N bond at 1580 cm^−1^ and the stretching vibration absorption peaks of C-N bond at 1145 and 990 cm^−1^, indicating that the framework of the Cu-Ag/g-C_3_N_4_/ZIF hybrid remained intact.

The successful formation of Cu-Ag bimetallic NPs along with the g-C_3_N_4_ nanosheet and ZIF-8 is confirmed from XRD analysis (Figure 2c,d). In Figure 2c, compared with the interlayer stacking diffraction peak (002) of bulk-C_3_N_4_, the 002 diffraction peak of the g-C_3_N_4_ nanosheet disappeared, indicating that the interlayer stacking of the block was destroyed into a single nanosheet. In Figure 2d, XRD analysis of the Cu-Ag/g-C_3_N_4_/ZIF hybrid showed the same characteristic diffraction peak as ZIF-8, namely (011), (002), (112), (013), (222), (114), (233), (134), (044), (244) and (235). It was proved that the structure of the hybrid was the framework of ZIF-8.

### 2.2. Cu-Ag/g-C_3_N_4_/ZIF Hybrid for the Detection of H_2_O_2_ and Glucose

The peroxidaselike activity of the Cu-Ag/g-C_3_N_4_/ZIF hybrid was demonstrated by investigating the catalytic oxidation reaction of substrate TMB to a blue 3,3′,5,5′-tetramethylbenzidine diimine (TMBDI) in the presence of H_2_O_2_. A comparison of peroxidaselike activity of 25 µL H_2_O_2_, 0.3 mg g-C_3_N_4_, 25 µL H_2_O_2_ and 0.3 mg g-C_3_N_4_, 0.3 mg Cu-Ag/ZIF, 25 µL H_2_O_2_ and 0.3 mg Cu-Ag/ZIF, 0.3 mg Cu-Ag/g-C_3_N_4_/ZIF; 25 µL H_2_O_2_ and 0.3 mg Cu-Ag/g-C_3_N_4_/ZIF to TMB is shown in terms of their UV−vis absorption spectra in Figure 3a. The presence of both H_2_O_2_ and catalyst is required for the oxidation of TMB to blue TMBDI. The results indicated that the catalytic performance of the Cu-Ag/g-C_3_N_4_/ZIF hybrid was significantly better than that of the g-C_3_N_4_ or Cu-Ag/ZIF hybrid. The optimal conditions for peroxidaselike activity of the Cu-Ag/g-C_3_N_4_/ZIF hybrid were determined considering different parameters such as reaction temperature, pH and concentration of TMB (Appendix A). The catalytic activity of the 0.3 mg Cu-Ag/g-C_3_N_4_/ZIF hybrid reached the maximum at 50 °C with the TMB of 500 µM and pH of 3.8.

In the presence of both H_2_O_2_ and the catalyst, the solution became blue and had significant absorption at 652 nm, indicating that the detection of H_2_O_2_ by the Cu-Ag/g-C_3_N_4_/ZIF hybrid was feasible. Figure 3b shows the absorbance at 652 nm against the concentration change of H_2_O_2_ from 0 to10 mM. The linear regression equation is A = 1.0389 C + 0.0078 (A is the absorbance; C is also the concentration of glucose in μM) with a linear range from 0.05 to 1 mM, and the limit of detection was calculated to be 2 μM (Figure 3c,d).

Because glucose produces gluconic acid and H_2_O_2_ in the acidic conditions in the presence of GOx, we designed a colorimetric method for the detection of glucose using a Cu-Ag/g-C_3_N_4_/ZIF hybrid. The result shows that the absorbance at 652 nm enhances significantly with the increasing concentrations of glucose from 0.1 to 2000 μM with an limit of detection LOD of 10 nM (Figure 4a,b). The linear regression equation is A = 0.0004 C + 0.02886 with a correlation coefficient of 0.991 (A is the absorbance; C is also the concentration of glucose in μM). Furthermore, the color variation is obvious upon visual observation (inset in Figure 4a). For comparison, the detection results of different nanozymes for glucose based on the colorimetric method are listed in Table 1.

### 2.3. The Selectivity of Cu-Ag/g-C_3_N_4_/ZIF Hybrid to Glucose Detection

Detection of glucose concentration was an important indicator of clinical diagnosis of glucose. We measured the absorbance response of three glucose analogues, (lactose, fructose and maltose) under the same experimental conditions. From Figure 5, the absorbance of the three glucose analogues at 652 nm was not significantly increased even at a high concentration of 5 mM, while the absorbance of glucose was significantly increased at a concentration of 0.5 mM. The experimental results showed that the Cu-Ag/g-C_3_N_4_/ZIF hybrid had better selectivity to glucose, although all the analogues have similar electron lone pairs and molecular sizes as glucose.

### 2.4. Colorimetric Detection of Glucose in Real Serum Samples

We extended our investigation towards the colorimetric detection of glucose by the Cu-Ag/g-C_3_N_4_/ZIF hybrid as the catalyst in the real serum samples of human volunteers by detecting the recoveries after adding a given amount of glucose. The performance of the Cu-Ag/g-C_3_N_4_/ZIF hybrid was comparable to that of a commercial glucometer (Table 2). The recovery was between 101.0% and 105.0%. These results suggest that this method is capable of glucose detection in human serum.

## 3. Experimental

### 3.1. Materials and Apparatus

All chemicals were at least analytical grade. Ultrapure water was used throughout the experiments. All solutions were freshly prepared before use. Zinc nitrate tetrahydrate (Zn(NO_3_)_2_·4H_2_O), copper nitrate trihydrate (Cu(NO_3_)_2_·3H_2_O), sodium borohydride (NaBH_4_), methanol (CH_3_OH), N-N-dimethylformamide (DMF),glucose, sulfuric acid (H_2_SO_4_) and hydrogen dioxide (H_2_O_2_) were purchased from Sinopharm Chemical Reagent Co., Ltd. (Shanghai, China). 2-methylimidazole (MeIM) and silver nitrate (AgNO_3_) were purchased from Aladdin Reagent Co., Ltd. (Shanghai, China). Melamine was purchased from McLean Co., Ltd. (Shanghai, China).

The Transmission electron microscopy (TEM) and High Resolution Transmission Electron Microscopy (HRTEM) images of the Cu-Ag/g-C_3_N_4_/ZIF hybrid were examined by FEI High Resolution Transmission Electron Microscopy (Thermo Fisher Scientific, Waltham, MA, USA). Scanning electron microscopy (SEM) images were prepared using a scanning electron microscope (JEOL JSM6510L, Tokyo, Japan). Fourier Transform Infra-Red (FTIR) spectra were taken with a spectrum on a FTIR spectrophotometer (Perkin-Elmer, Waltham, MA, USA). Anode rotating target X-ray diffraction (XRD, Bruker D 8 Advance, Bragg-Brentano, Germany) was carried out to identify crystal structure. The powders deposited on a silicon zero-background sample holder were scanned at a rate of 0.02° (2θ) per second over the range of 5°–60° (2θ). UV-vis absorption spectra were measured at room temperature with a Cary 50 UV-vis spectrophotometer (UV-vis, Shimadzu, Japan) over the range of 200–550 nm. X-ray photoelectron spectroscopy (XPS, Thermofisher, Hillsboro, OR, USA) measurements were performed using a Thermo Scientific K-Alpha spectrometer.

### 3.2. One-Step Method to Synthesize Cu-Ag/g-C_3_N_4_/ZIF Hybrid

The bulk g-C_3_N_4_ was prepared by direct calcination from melamine: 5 g of melamine was placed in an alumina crucible with a cover and heated from room temperature to 535 °C for 3h, then cooled down naturally. The obtained yellow bulk solid was milled with a quartz mortar and collected. The bulk g-C_3_N_4_ was modified via Hummers [31]: Under the condition of ice water bath, 10 g g-C_3_N_4_ and 230 mL H_2_SO_4_ (98%) were stirred and mixed well. An amount of 30 g KMnO_4_ was slowly added and stirred continuously for 30 min. The mixture was transferred to 1.4 L ultrapure water, then 100 mL H_2_O_2_ (30%) was added. G-C_3_N_4_ was obtained by centrifugation after standing for 24 h. Amounts of 70 mg ZIF-8 and 30 mg g-C_3_N_4_ were dispersed in 20 mL methanol solution under ultrasonic condition for 5 min. Amounts of 0.4 mL Cu(NO_3_)_2_ (0.0906 g, 0.15 M) and 0.2 mL AgNO_3_ (0.0378 g) were added under the protection of N_2_. An amount of 2 mL freshly prepared NaBH_4_ solution (0.0743 g, 1 M) was injected during continuously stirring. The Cu-Ag/g-C_3_N_4_/ZIF hybrids were obtained via centrifugation and washed with methanol for three times. The catalyst was dried under vacuum for 5 h, and freezed for preservation.

### 3.3. Cu-Ag/g-C_3_N_4_/ZIF Hybrid for Colorimetric Detection of H_2_O_2_ and Glucose

H_2_O_2_ detection was performed at 50 °C with pH of 3.5, adjusted with acetate buffer solution. An amount of 0.3 mg catalyst was added to 0.5 nM TMB aqueous solution. The pH was adjusted to 3.8 by acetate buffer solution, making the total volume 2 mL for UV detection. The different concentrations of H_2_O_2_ were added into the above solution. After reaction for 20 min in a 50 °C water bath, the absorbance change of the solution at 652 nm was measured, obtaining a linear range of detection of H_2_O_2_ by the Cu-Ag/g-C_3_N_4_/ZIF hybrid.

The detection of glucose was performed with the following steps: Firstly, 50 µL 1 mg·mL^−1^ glucose oxidase and different concentrations of glucose were incubated at 37 °C with a pH of 7 for 1 h to obtain different concentrations of H_2_O_2_ solution. Then, 0.5 mL 2 mM TMB and 0.3 mg catalyst were added into the above solution. Finally, the pH was adjusted to 3.8 by the acetate buffer solution to make a total volume of 2 mL for UV detection. After reaction for 20 min in a 50 °C water bath, the absorbance change of the solution at 652 nm was measured, obtaining a linear range of detection of glucose by the Cu-Ag/g-C_3_N_4_/ZIF hybrid.

## 4. Conclusions

In conclusion, the Cu-Ag/g-C_3_N_4_/ZIF hybrid was synthesized and characterized by Transmission electron microscopy (TEM), Fourier transformed infrared (FTIR), the X-ray diffraction (XRD) and X-ray photoelectron spectroscopy (XPS). The hybrid showed peroxidaselike activity to catalyze the oxidation of TMB to TMBDI in the presence of H_2_O_2_. A facile colorimetric method for detection of the glucose was established by combining the peroxidaselike catalytic activity of the Cu-Ag/g-C_3_N_4_/ZIF hybrid with glucose oxidase. This method was applied for detecting glucose in human serum, owing to the significant potential for the detection of metabolites with H_2_O_2_-generation reactions.

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
