# Peer review of "A Colorimetric Assay for the Detection of Glucose and H2O2 Based on Cu-Ag/g-C3N4/ZIF Hybrids with Superior Peroxidase Mimetic Activity"

_molecules, 2020, doi:10.3390/molecules25194432_

Round 1
Reviewer 1 Report
This is paper that describes the synthesis and characterization of Cu-Ag/g-C3N4/ZIF nanoparticles. The new species have been characterized by a variety of techniques including TEM, IR, XRD and XPS. They also possess very good performance for glucose detection. This is an interesting piece of work and I would recommend its publication in Molecules after the authors have addressed the below comments:
1. There are several typographic errors throughout the manuscript and careful editing is required before publication.
2. In the experimental section, the authors report that they performed tga studies and gas measurements in the nanoparticles. However, the discussion of the results from these measurements are not included in the main part. It would be great to see how those properties for the hybrids are compared to the corresponding properties of the the pristine ZIF MOF.
Reviewer 2 Report
The authors report the synthesis of Cu-Ag bimetallic nanopartiles and g-C3N4
nanosheets decorated on zeolitic imidazolate framework-8 (ZIF-8) to form Cu-Ag/g-C3N4/ZIF hybrid. The hybrid was synthesized and characterized by Transmission electron microscopy (TEM), Fourier transformed infrared(FTIR), the X-ray diffraction (XRD), and X-ray photoelectron spectroscopy(XPS). The Cu-Ag/g-C3N4/ZIF hybrid has intrinsic peroxidase-like catalytic activity
towards the oxidation of TMB in presence of H2O2.
There are some points in the manuscript that should be reconsidered:
01 Fig 2, FTIR spectra: relative transmittance should be on y-ax. Please record the spectra once again since the intensities of the samples on the left Fig are very low.
02 XRD spectra of g-C3N4 has very low quality, peaks have low intensities.
03 Chapter 2.2 says The presence of both H2O2 and catalyst is required for the oxidation of TMB to blue TMBDI. What happens if there is no oxidant or/and no catalyst?
04 Table 2 has the recovery data. Please explain how is it calculated and what means recovery >100 %.
05 Experimental 4.1 has data for TG experiments although TG analysis has not been presented and discussed in the manuscript
06 Experimental 4.2 Please explain why is bulk g-C3N4 prepared from melamine heated to 535 °C.
Due to the above reasons, I suggest a major revision.
Round 2
Reviewer 2 Report
The authors provided all the asked corrections and I agree that the manuscript proceeds for publishing.